# Genome-Wide Mining of the Tandem Duplicated Type III Polyketide Synthases and Their Expression, Structure Analysis of *Senna tora*

**DOI:** 10.3390/ijms24054837

**Published:** 2023-03-02

**Authors:** Zeping Cai, Xingkun Zhao, Chaoye Zhou, Ting Fang, Guodao Liu, Jiajia Luo

**Affiliations:** 1Key Laboratory of Genetics and Germplasm Innovation of Tropical Special Forest Trees and Ornamental Plants, Ministry of Education, College of Forestry, Hainan University, Haikou 570228, China; 2College of Tropical Crops & College of Life Sciences, Hainan University, Haikou 570228, China; 3Tropical Crops Genetic Resources Institute, Chinese Academy of Tropical Agricultural Sciences, Haikou 571101, China

**Keywords:** chalcone synthase-like gene, expression and structure analysis, *Senna tora*, tandem duplication, type III polyketide synthase

## Abstract

*Senna tora* is one of the homologous crops used as a medicinal food containing an abundance of anthraquinones. Type III polyketide synthases (PKSs) are key enzymes that catalyze polyketide formation; in particular, the chalcone synthase-like (CHS-L) genes are involved in anthraquinone production. Tandem duplication is a fundamental mechanism for gene family expansion. However, the analysis of the tandem duplicated genes (TDGs) and the identification and characterization of PKSs have not been reported for *S. tora*. Herein, we identified 3087 TDGs in the *S. tora* genome; the synonymous substitution rates (Ks) analysis indicated that the TDGs had recently undergone duplication. The Kyoto Encyclopedia of Genes and Genomes (KEGG) enrichment analysis showed that the type III *PKSs* were the most enriched TDGs involved in the biosynthesis of the secondary metabolite pathways, as evidenced by 14 tandem duplicated *CHS-L* genes. Subsequently, we identified 30 type III *PKSs* with complete sequences in the *S. tora* genome. Based on the phylogenetic analysis, the type III PKSs were classified into three groups. The protein conserved motifs and key active residues showed similar patterns in the same group. The transcriptome analysis showed that the chalcone synthase (CHS) genes were more highly expressed in the leaves than in the seeds in *S. tora*. The transcriptome and qRT-PCR analysis showed that the *CHS-L* genes had a higher expression in the seeds than in other tissues, particularly seven tandem duplicated *CHS-L2*/*3*/*5*/*6*/*9*/*10*/*13* genes. The key active-site residues and three-dimensional models of the CHS-L2/3/5/6/9/10/13 proteins showed slight variation. These results indicated that the rich anthraquinones in *S. tora* seeds might be ascribed to the *PKSs’* expansion from tandem duplication, and the seven key *CHS-L2*/*3*/*5*/*6*/*9*/*10*/*13* genes provide candidate genes for further research. Our study provides an important basis for further research on the regulation of anthraquinones’ biosynthesis in *S. tora*.

## 1. Introduction

*Senna tora* is a leguminous annual plant that is widely cultivated and mainly used as a medicinal food source throughout the tropical and subtropical parts of Asian and African countries [1]. The mature seeds of this plant together with another species, *Senna obtusifolia*, were first introduced as the traditional Chinese medicine Jue-ming-zi in the Chinese herbal medicine book “Shen nong ben cao jing” [2]. Jue-ming-zi is popularly served as a healthy roasted tea throughout Korea and China [3]. The leaves and seeds of *S. tora* have long been used as food ingredients for animals and human [1]. Importantly, it Is also valued as a traditional medicine to treat different diseases, for example, skin diseases (leprosy and ringworm), gastrointestinal disorders (flatulence, dyspepsia, intestinal dryness, etc.), inflammation (bronchitis and cough), and vision problems (xerophthalmia) [4]. The medicinal value of *S. tora* is mainly attributed to its various bioactive compounds, such as anthraquinones, flavonoids, alkaloids, and terpenoids, that have antioxidant, anti-inflammatory, and antibacterial properties [5,6]. *S. tora* leaves were found to contain phenols and flavonoids that had a significant antioxidant activity [4]. Anthraquinones are the main active ingredients of *S. tora* seeds, especially aurantio-obtusin, a specific component found in *Senna* plants; hence, *S. tora* was considered as a model plant to research the biosynthetic pathways of anthraquinones.

Flavonoids and anthraquinones, polyketone secondary metabolites, are important bioactive compounds of plants. Type III polyketide syntheases (PKSs) are specialized enzymes that catalyze the decarboxylative condensation of simple two-carbon acetate units (typically derived from malonyl-CoA) with an acyl starter to form polyketides [7]. Chalcone synthases (CHSs) are the most extensively studied and the most prevalent members of type III PKSs. They catalyze the action of the three malonyl-CoA molecules, which are sequentially condensed with a starter substrate ρ-coumaroyl-CoA and produce naringenin chalcone [8]. The CHSs have been well characterized as the key enzymes of flavonoid biosynthesis in various plant species, such as *Medicago sativa*, *Arabidopsis thaliana*, and *Gossypium barbadense* [8,9,10]. Recently, reports speculated that plants made anthraquinones via type III PKSs in the polyketide pathway [11,12]. A few previous studies indicated that the type III PKS enzymes could catalyze seven successive condensations of malonyl-CoA to produce linear octaketides, subsequently undergoing cyclization, decarboxylation, and dehydration reactions to produce anthrones, such as the PKS4/PKS5/OKS of *A. arborescens* [13], the OKS of *C. bicapsularis* [14], and the chalcone synthase-like 9 (CHS-L9) of *S. tora* [15]. However, the detailed regulation mechanisms of anthraquinone biosynthesis via the polyketide pathway in plants are largely unknown.

Gene duplication acts as the primary driver of gene family expansion, which occurs at various levels through whole genome duplication and single gene duplication [16]. Tandem duplication (TD), a typical type of single-gene duplication, produces tandem duplicated genes (TDGs) in the same genomic neighborhood [17]. The TD of genes has been found to be important for the plant formation of phenotypic properties and the adaptation to environmental stresses [18]. For example, tandem and segmental duplication of genes contributed to disease resistance in *A. thanlian* [19]. Liu et al. (2021) reported that 21 out of 59 glutathione S-transferase (*GST*) genes and 14 out of 18 2,4-dihydroxy-1,4-benzoxazin-3-one-glucoside dioxygenase (*BX6*) genes were increased by TD, which might be important for the strong biotic and abiotic tolerance in *Cajanus cajan* [20]. With more and more whole genome sequences of plant species being reported, the potential contributions and possible scenarios of gene duplication have gradually become a focus. At present, little is known about the TDGs and their contributions to the rich anthraquinones of *S. tora*.

In this work, we identified the TDGs in *S. tora* at the genome-wide level, and we report the global distribution, the synonymous substitution rates (Ks), and the enrichment analysis of the Kyoto Encyclopedia of Genes and Genomes (KEGG) pathways of TDGs. Furthermore, type III PKSs, the most abundant TDGs significantly enriched in the KEGG pathways, were further analyzed, including the identification of family members, the comparison of the active sites of the putative protein sequences, the construction of the phylogenetic tree, and the analysis of the expression patterns. Finally, we screened seven tandem duplicated *CHS-L* genes that were considered as key genes responsible for the biosynthesis of anthraquinones. The three-dimensional model building and expression level detection of these *CHS-L* genes were subsequently performed. This study lays a foundation for further research on the molecular mechanisms of anthraquinone biosynthesis.

## 2. Results

### 2.1. Genomic Distribution, Ks and KEGG Enrichment Analysis of the Tandem Duplicated Genes

The genomic sequence of *S. tora* consisted of a 526 Mb DNA sequence, which was assembled into 13 chromosomes, and 45,268 protein-coding genes were annotated. A total of 3087 TDGs (6.82% of the predicted genes) were identified in the genome, and 3065 TDGs were relatively evenly distributed on 13 pseudochromosomes, with the highest number on Chr09, including 325 TDGs (Appendix A). Interestingly, the location density showed that the TDGs were mainly concentrated on both ends of pseudochromosomes, typically Chr01, Chr02, Chr06, Chr07, and Chr10 (Figure 1a). In addition, the Ks of 1767 TDG pairs were calculated (Appendix A), and the Ks density distribution showed that a single peak was displayed at 0.10−0.14, which indicated that these TDGs had undergone duplication recently (Figure 1b).

The results of the TDGs’ KEGG enrichment analysis showed that eight categories were significantly enriched with *Padj* < 0.05, including 19 pathway terms (Figure 1c, Appendix A). The group of “biosynthesis of other secondary metabolites” had the most pathway terms, including flavonoid biosynthesis, indole alkaloid biosynthesis, isoflavonoid biosynthesis, etc., with eight terms (Figure 1c). In the group of “biosynthesis of other secondary metabolites”, we found that 175 TDGs were enriched. The gene family classification showed that the top eight families (TDGs ≥ 6) included “Chalcone synthase”, “UDP-glycosyltransferase”, “Beta-glucosidase”, “Cinnamoyl-CoA reductase”, etc., and the “Chalcone synthase” family contained the most TDGs, with 19 (Figure 1d, Appendix A).

### 2.2. The Identification, Distribution, and Protein Phylogenetic Tree Analysis of the Type III Polyketide Synthases

A species phylogenetic tree including *S. tora* with the other 12 plant species showed that *S. tora* had the closest relationship to *Chamaecrista fasciculata* from the Caesalipinoideae clade (Figure 2a). The identification of type III *PKSs* in 13 plant species indicated that the number was conservative; the CHS family genes ranged from 6 (*Lupinus albus*) to 35 (*Arachis duranensis*), and the number of CHS genes was higher than 30 in four species, including *A. duranensis* (35 genes), *Arachis ipaensis* (34 genes), *C. fasciculata* (33 genes), and *S. tora* (30 genes) (Figure 2a, Appendix A).

In the genome of *S. tora*, 33 type III *PKSs* were identified, but three genes (G2W53_001187, G2W53_001804, and G2W53_022828) consisted of incomplete domains (PF00159 and PF02797), while the other 30 genes with the complete domains had two exons and one intron, with the second exon being longer than the first (Figure 2b). In addition, 20 out of 33 type III *PKSs* were derived from tandem duplication in *S. tora*, including five TDGs on Chr03 and 15 TDGs on Chr07 (Figure 2b). Furthermore, the Ks values of these 14 TDG pairs ranged from 0.085 (G2W53_0022837 vs. G2W53_022838) to 0.841 (G2W53_0022827 vs. G2W53_022828). A total of eight TDG pairs had a value less than 0.3, which indicated relatively recent duplications (Figure 2b, Appendix A).

The 32 type III PKS superfamily members (G2W53_022828 was eliminated) were divided into 5 PKS, 16 CHS-L, and 14 CHS family genes; we found that 14 out of 16 CHS-L and 5 out of 14 CHS proteins were tandem duplications (Figure 2b,c). A phylogenetic tree of the type III PKS proteins showed that 14 tandem duplicated CHS-L and five tandem duplicated CHS proteins of *S. tora* were clustered as independent branches without other plant genes, respectively (Figure 2c).

### 2.3. Motif Analysis of the Type III Polyketide Synthases

The protein molecular masses of 30 type III PKSs with complete domains were predicted, and we found that the protein molecular masses were extremely similar, ranging from 41.95 kDa (G2W53_002718) to 44.91 kDa (G2W53_024354) (Figure 3). A total of 15 conserved motifs were identified in the 30 type III PKS predicted proteins. We found that the PKS members in the same family exhibited similar motif patterns (Figure 3). Five motifs (motifs 1, 2, 5, 6, and 10) were present in all members, suggesting that these motifs might be involved in their common function. The CHS and CHS-L family members exhibited an extremely similar motif distribution; they contained nine motifs at the same time. However, a few motifs diverged between the CHS and CHS-L family members. For example, motifs 12 and 13 were found only in the CHS family members, motifs 7 and 14 were only present in the CHS-L family members, and motif 9 was present in the CHS and PKS family members (Figure 3). These specific motifs in different families might be responsible for their divergent functions.

### 2.4. Protein Active-Site Analysis of the Type III Polyketide Synthases

Comparing the active-site residues of 30 type III PKSs, except for the three members with incomplete domains, the sequence analysis revealed that the catalytic triad of Cys164, His303, and Asn336 was conserved in all members (Figure 4). In addition, the critical active-site residue Thr197 and the residues lining the active site (Phe215, Gly256, Phe265, and Ser338), as well as other critical residues for the CoA binding in the CHS family members, were also conserved (Figure 4). However, for the CHS-L family members, the active-site residues were varied. For example, the critical residue Thr197 in CHS was replaced by Asn/Ser, the residues corresponding to Gly256, Phe265, and Ser338 in the CHS were also substituted with Leu, Tyr, and Trp in the CHS-L, respectively. The CoA binding residues in the CHS-L family differed not only among the members but also from the CHS family (Figure 4). Finally, the active-site residues of the PKS in *S. tora* or *A. thaliana* were conserved, but some residues differed from those of the CHS-L or CHS. Notably, the residue Thr197 of the CHS was replaced by Gly in the PKS (Figure 4).

### 2.5. Transcriptome Expression Analysis of the Type III Polyketide Synthases

We reanalyzed the RNA-Seq data of *Senna* (*S. tora* and *S. obtusifolia*) to detect the relative expression levels of the type III PKS genes in the leaves and seeds (Appendix A). The results showed that the *CHS* family genes were mainly expressed in the leaves, reflected by 8 out of 12 *CHS* genes with relatively higher expression levels in the leaves than the seeds (Figure 5). In contrast, the *CHS-L* family genes were mainly expressed in the seeds. Markedly, we observed that seven *CHS-L* genes (including *CHS-L2/3/5/6/9/10/13*) had higher expression levels in the seeds than the leaves (Figure 5). However, the *PKS1* and *PKS2* genes had unclear expression tendencies (Figure 5).

### 2.6. Quantitative Real-Time PCR of the Chalcone Synthases-like Genes

The transcriptome analysis showed that seven *CHS-L* genes had higher expression levels in the seeds than the leaves (Figure 5). Subsequently, we identified the six CHS-L genes’ (*CHS-L2/5/6/9/10/13*) expression levels in different tissues by qRT-PCR, except for *CHS-L3* with relatively low expression levels in the seeds’ transcriptome (Figure 6). The results showed that six *CHS-L* genes had extremely low expression levels in the Senna leaves and pods, and there was no significant difference between the two tissues (Figure 6b). However, the six *CHS-L* genes had significantly higher expression levels in the seeds than the leaves and pods (*p* < 0.05), and the highest expression levels were presented in the development stage 2 seeds (Figure 6). The results suggested that these *CHS-L* genes were mainly expressed in seeds, which was consistent with the transcriptome data.

### 2.7. Structure Analysis of the Chalcone Synthases-like Genes

We compared and analyzed the active-site residues and three-dimensional structures of seven CHS-L proteins (Figure 7). A CoA binding residue (Arg) of the CHS-L5 protein was different from the other CHS-L proteins (Lys). The CHS-L6 protein had two critical residues that were different from the others; the Asn197 in CHS-L6 was replaced by Ser197, and a CoA binding residue (Ser) in CHS-L6 was substituted with Ala (Figure 7a). The CHS-L2/3/9/10/13 proteins had the same active-site residues. Furthermore, homology modeling of these seven CHS-L proteins showed that some active-site residues of CHS-L5/6 were different from the CHS-L2/3/9/10/13 proteins (Figure 7b). We found that the area, volume, and length of the CHS-L6 predicted pocket were smaller or shorter than the other CHS-L proteins. The length of the CHS-L5 predicted pocket was shorter than the CHS-L/2/3/9/10/13 proteins (Figure 7b).

## 3. Discussion

*S. tora* is a widely cultivated and used medicinal and food crop in subtropical and tropical countries, especially India, China, Sri Lanka, Nepal, and the Korean Peninsula [12]. The medicinal value of *S. tora* is attributed to the presence of secondary metabolites, especially flavonoids and anthraquinones. Previous studies have focused on several areas of *S. tora* genomics, as reflected by the transcriptome analysis, chloroplast genome assembly, and chromosome dynamics evaluation [3,21]. However, studies on the genome-wide analysis of TDGs and the identification of the type III *PKS* genes involved in flavonoid and anthraquinone biosynthesis in *S. tora* have not been reported.

Recently, Kang et al. [15] released the genome sequences of *S. tora*, which were found to contain 3371 expanded gene families, but they did not present a recent whole-genome duplication (WGD) event. In this study, we identified a total of 3087 TDGs in the *S. tora* genome, of which 3065 were distributed in 13 pseudochromosomes with higher density on two ends of chromosomes (Figure 1a). We further found that the TDGs of *S. tora* resulted from recent gene expansion, reflected by the Ks’ occurring peak at 0.10–0.14 (Figure 1b). In addition, the KEGG enrichment analysis showed that the TDGs were significantly enriched in the secondary metabolite biosynthesis pathways, including flavonoids, phenylpropanoids, polyketides, etc., (Figure 1c). This was consistent with the KEGG terms enriched by the expansion gene families of the *S. tora* genome, as reported by Kang et al. [15]. Moreover, we further dissected the 175 TDGs involved in the eight pathways of the “biosynthesis of other secondary metabolites” category, in which 19 type III *PKS* TDGs were contained (Figure 1c,d). Previous studies have reported that the type III PKSs were key enzymes involved in polyketides’ production [22,23]. Taken together, those results indicated that the recent TD events might contribute to the expansion of gene families in the *S. tora* genome, and the TD expansion of the type III *PKSs* might be important for the abundance of flavonoids and polyketides in *S. tora*.

Type III *PKSs* mainly consist of the typical *CHS* and chalcone synthase-like (*CHS-L*) genes [23]. The *CHS* genes have been demonstrated to participate in flavonoid biosynthesis, and the *CHS-L* genes were proposed to produce some other-specific polyketides [24]. In this study, we identified 33 type III *PKSs* in the *S. tora* genome, including 13 *CHS* genes and 16 *CHS-L* genes (Figure 2). Furthermore, we found that 14 out of 16 *CHS-L* genes were tandem duplicated, and 8 out of 11 tandem duplicated *CHS-L* gene pairs were recently occurring (Ks < 0.3) (Figure 2). A previous study reported that the CHS-L gene family has rapidly expanded in the *S. tora* genome [15]. Thus, we suggest that tandem duplication is the driving force for the CHS-L gene family expansion evolution in *S. tora*.

The protein phylogenetic tree analysis displayed that the CHSs of *S. tora* clustered with MsCHS2 and AtCHS, which are involved in flavonoid production [8,9], but the CHS-Ls of *S. tora* were grouped into an independent cluster (Figure 4). The structure of the type III PKS active sites is believed to provide clues as to their substrate specificity and catalytic mechanism [25]. The comparation of the active sites showed that the CHSs of *S. tora* harbored perfectly consistent critical active sites (Figure 4); this was similar to prior studies [13,25,26]. Although the catalytic triad (Cys164-His303-Asn336) of the CHS-Ls of *S. tora* was conserved with other type III PKSs, the other critical active sites fluctuated greatly (Figure 4); this might play an important role in producing a variety of anthraquinones in *S. tora*.

Transcriptome analysis showed that the *CHS* and *CHS-L* genes were highly expressed in the leaves and seeds of *S. tora*, respectively (Figure 5). This corresponded to the rich accumulations of the leaf flavonoids and seed anthraquinones in *S. tora* [2,5]. Subsequently, we screened six key *CHS-L* genes for qRT-PCR verification, and the results also demonstrated that these *CHS-L* genes had higher expression levels in *S. tora* seeds than in the leaves and pods (Figure 6). Previous studies revealed that the functional diversity of type III PKSs was principally derived from different active cavity sites, and even the single 197-site residue could determine the number of malonyl-CoA condensations [13,25,27]. Therefore, we compared the key active sites of seven CHS-L genes (*CHS-L2*/*3*/*5*/*6*/*9*/*10*/*13*) that were highly expressed in the *S. tora* seeds. We found that the CHS-L6 harbored different 197-site residues and one key active site from the other six CHS-L genes; one key active site of CHS-L5 also was different, but the five CHS-L members (CHS-L2/3/9/10/13) had the same key active sites (Figure 7). Furthermore, studies have shown that the size of the type III PKSs active cavities was closely related to the size of the polyketides, as the size of the active cavity was generally proportional to the number of cyclization events [7,25]. In this study, the smaller size of the CHS-L6 cavity suggests that it might produce a small polyketide (Figure 7). Altogether, we screened seven *CHS-L2/3/5/6/9/10/13* genes that were potential candidates for anthraquinone biosynthesis, which will provide a theoretical basis for the further study of the molecular mechanism of anthraquinone synthesis in *S. tora*.

## 4. Materials and Methods

### 4.1. Data Sources

The genomic sequences, gene sequences, protein sequences, and the general feature format (GFF) files of *Senna* (*S. tora* cv. Myeongyun) were downloaded from the National Agricultural Biotechnology Information Center (NABIC, http://nabic.rda.go.kr/Species/Senna_tora2, accessed on 22 September 2022), according to Kang et al. [15]. The RNA-Seq data of *Senna* (*S. tora*) leaves, young seeds, and mature seeds were downloaded from the National Center for Biotechnology Information (NCBI) Sequence Read Archive (SRA) database (accession number SRP159435), as described by Kang et al. [12]. The RNA-Seq data of *Senna* (*S. obtusifolia*) leaves and seeds were also obtained from the SRA database (accession number SRP144670) of NCBI [1]. The genomic data of the other 11 legume species (*Phaseolus vulgaris*, *Glycine max*, *Cajanus cajan*, *Medicago truncatula*, *Cicer arietinum*, *Lupinus albus*, *Arachis duranensis*, *Arachis ipaensis*, *Chamaecrista fasciculata*, *Mimosa pudica*, and *Cercis canadensis*) and an outlier (*Vitis vinifera*) were obtained from the Phytozome (version 13) database (https://phytozome-next.jgi.doe.gov/, accessed on 24 September 2022).

### 4.2. Genome-Wide Identification and Analysis of the Tandem Duplicated Genes

The TDGs of *Senna* were identified in the genome by MCScanX software according to the method of Liu et al. [20]. In brief, BLASTP (version 2.9.0) software was used to identify the protein sequences with an E-value of <1 × 10^−10^. The blast results were analyzed to identify the duplicated gene pairs by the duplicate_gene_classifier tool of the MCScanX package, according to the following criteria: the genes were orthologous, located at a distance within 100 kb, and separated by no or a few nonhomologous intervening ‘spacer’ genes.

For the synonymous substitution rates (Ks) and the KEGG pathway enrichment analysis of TDGs, the Ks values of 1767 tandem duplicated gene (TDG) pairs in *Senna* were calculated by TBtools (version 1.098669) with NG_method [28], and the Ks distribution visualization was performed using the ggplots package in R software (version 4.1.0). In addition, the KEGG pathway functional enrichment of the *S. tora* TDGs was analyzed by phyper in the R platform (version 4.1.0). The subsequent *P*-value and *Padj* were calculated with the hypergeometric test and the qvalue package (version 3.10), respectively. The KEGG pathways with *Padj* < 0.05 were considered to be significantly enriched.

### 4.3. Identification of the Type III Polyketide Synthase Genes

The type III polyketide synthase genes of *Senna* and the other 12 plant species were identified by searches of the pfam domains, as described by Hu et al. [29]. Firstly, the type III polyketide synthase superfamily conserved domains were downloaded from the Pfam database (http://pfam.xfam.org/, accessed on 2 October 2022), including Chal_sti_synt_N (PF00195) and Chal_sti_synt_C (PF02797). Hmmer (version 3.3.2) was used to search for the domains in the protein sequences with an E-value < 1 × 10^-5^. The genes with two conserved domains were considered candidate genes, further confirmed by Pfam (http://pfam.xfam.org/search, accessed on 3 October 2022).

### 4.4. Characteristics of the Type III Polyketide Synthase Genes

The distribution of the type III *PKSs* on the chromosome was visualized using MG2C (version 2.1, http://mg2c.iask.in/mg2c_v2.1/, accessed on 9 October 2022, Chinese Academy of Agricultural Sciences, Beijing, China). The gene structure was displayed by GSDS (version 2.0, http://gsds.gao-lab.org/, accessed on 10 October 2022). A phylogenetic tree of the type III polyketide synthase predicted protein sequences from *Senna*, *G. max*, *M. truncatula*, and *Arabidopsis thaliana* was produced according to the method described by Hu et al. [29]. Multiple sequence alignment was performed with ClustalW. The phylogenetic tree was constructed with MEGA (version 7.0.26) using the neighbor-joining method with 1000 bootstrap replicates. An online tool iTOL (version 5, https://itol.embl.de/, accessed on 10 October 2022, European Molecular Biology Laboratory, Heidelberg, Baden-Württemberg, Germany) was used to display and annotate the phylogenetic tree. The conserved motifs of 30 type III PKS amino acid sequences with complete domains were identified by the Multiple Expectation Maximization for Motif Elicitation (MEME, version 5.5.1, http://meme-suite.org, accessed on 11 October 2022) online tool as described by Xiao et al. [30]. The parameters of the maximum number and the optimum width of the motifs were set to 15 and 6–50, respectively.

### 4.5. Transcriptome Expression Analysis of the Type III Polyketide Synthase Genes

To unravel the expression profile of the type III polyketide synthase genes in the leaves and seeds, the transcriptome data of *Senna* (*S. tora* and *S. obtusifolia*) were reanalyzed, and the genomic assembly was used as a reference [1,12,15]. Clean data were obtained by removing the adaptor and low-quality reads using the fastp (version 0.12.4) software with -q 20 -u 30 -n 15 parameters. For transcript abundance calculation, the sequencing reads were mapped onto the genomic sequences using hisat2 (version 2.1.0) with the default parameters, and all alignment rates were above 95% (Appendix A). The reads count was calculated by the FeatureCounts software, which is incorporated into the subread (version 1.6.4) package. The gene expression was normalized to transcripts per million (TPM) by the R platform, depending on the GenomicFeatures (version 1.44.2) and dplyr (version 1.0.7) packages.

### 4.6. Quantitative Real-Time PCR of the Chalcone Synthase-Like Genes

To analyze the expression levels of six *CHS-L* genes (*CHS-L2/5/6/9/10/13*) related to anthraquinone biosynthesis in the leaves, pods, and seeds of *S. tora*, the total RNA was extracted and subjected to quantitative real-time PCR (qRT-PCR) analysis. The *CHS-L3* gene with relatively low expression in the tissues was excluded from further qRT-PCR analysis. Following the manufacturer’s instructions, the total RNA was extracted using TRNZol reagent (Cat# DP424, Tiangen Biotech, Beijing, China), the cDNA synthesis was performed using a HiScript II 1st Strand Cdna Synthesis Kit plus Gdna eraser (Cat# R212-02, Vazyme, Nanjing, China), and the qRT-PCR was performed using ChamQ Universal SYBR qPCR Master Mix (SYBR Green, Cat# Q711-02, Vazyme, Nanjing, China) in a QuantStudio 6 Flex qRT-PCR system (ABI QuantStudio 6, Applied Biosystems, Waltham, MA, USA). Subsequently, we evaluated the expression stability of 42 commonly used internal control genes at transcriptome levels according to previous studies [31,32], and the *PTB* gene (G2W53_013008) was the most stable one (Appendix A; Appendix A). Thus, the relative expression levels of the *CHS-L* genes were calculated by the 2^−ΔΔCT^ algorithm using the PTB as the internal control, as described by Li et al. [33]. All gene sequences and primers used for the qRT-PCR analysis are listed in Appendix A.

### 4.7. Three-Dimensional Structure Analysis of the Chalcone Synthase-Like Genes

The homology models of seven CHS-L genes (CHS-L2/3/5/6/9/10/13) with higher expression levels in *Senna* seeds were generated using the previously described method with some modification [7]. The initial model building was performed by the SWISS-MODEL package (https://swissmodel.expasy.org/, accessed on 20 October 2022, University of Basel, Basel, Canton of Basel-Stadt, Switzerland), based on the monomer of the A-chain modeling pattern. Then, the quality of the models was checked with the PROCHECK (version 3.5) program in the SAVES (version 6.0) platform (https://saves.mbi.ucla.edu/, accessed on 20 October 2022) [34]. Details of the modeling and quality assessment are displayed in Appendix A. Above 92.4% of the residues in 7 CHS-L proteins were in the most favored regions of the Ramachandran plot, 5.6−7.1% in the additional allowed regions, 0.3−0.9% in the generously allowed regions, and 0.0−0.6% in the disallowed regions (Appendix A). The surface and cavity volume of the solvent-accessible pockets were identified by the webserver CASTp (version 3.0, http://cast.engr.uic.edu/cast/, accessed on 25 October 2022). All crystallographic figures were prepared with the program PyMOL (version 2.3.2, https://www.pymol.org, accessed on 25 October 2022, Schrödinger, Columbia, South Carolina, USA).

## 5. Conclusions

In this study, a total of 3087 TDGs in the *S. tora* genome was identified, and 99.29% (2065) were located on 13 pseudochromosomes. The chromosomal distribution, Ks calculation, and KEGG enrichment analysis of TDGs were performed. Furthermore, the type III PKSs, the most enriched gene family involved in the secondary metabolite pathway of KEGG analysis, were subjected to genome-wide identification, conserved domains and motifs analysis, protein phylogenetic tree construction, proteins key active-sites comparison, and transcriptome expression pattern analysis. Finally, we further analyzed the tissue expression pattern and protein three-dimensional structure of seven key *CHS-L* genes (*CHS-L2*/*3*/*5/6/9/10/13*), which might be involved in anthraquinones producing. The study suggests that the *CHS-L* genes recently expanded by tandem duplication exhibited preferential expressions in the seeds and showed slight protein active site and structural variation in *S. tora*. In conclusion, this study will provide helpful information for further functional analysis of the *CHS-L* genes in the regulation of anthraquinone biosynthesis in *S. tora*.

## Figures and Tables

**Figure 1 ijms-24-04837-f001:**
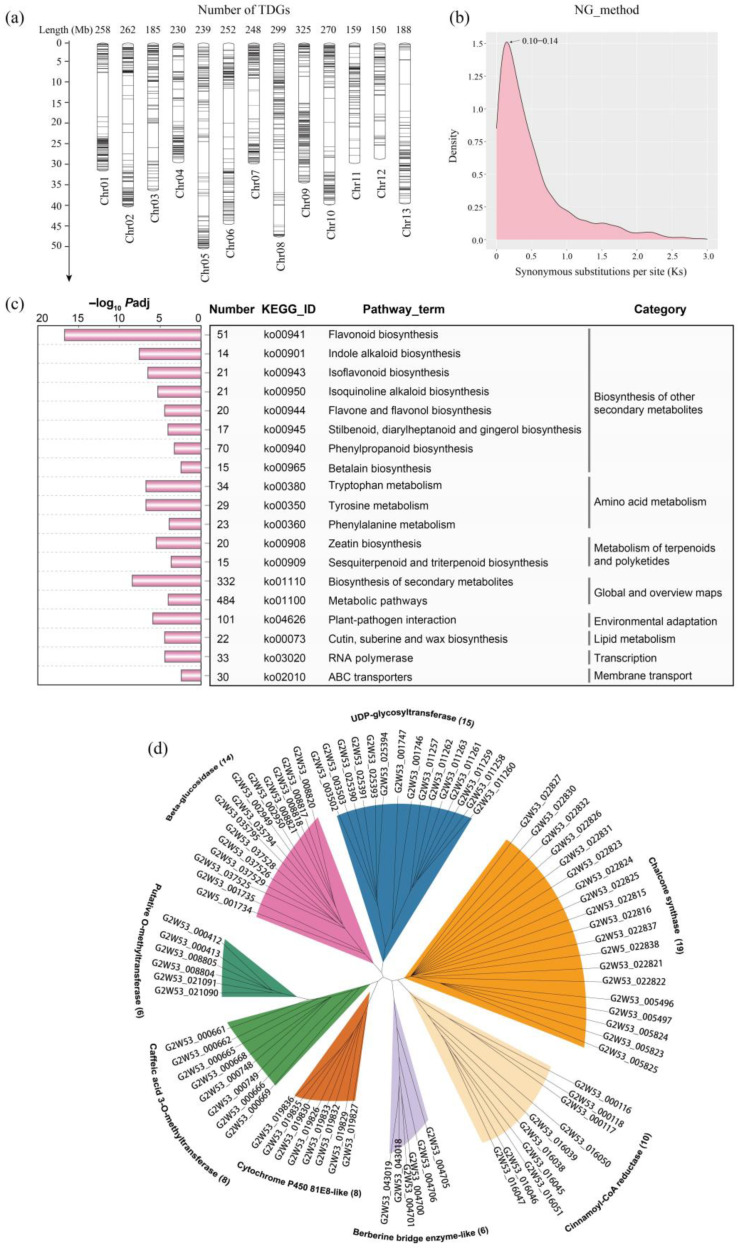
Genome-wise analysis of the tandem duplicated genes (TDGs) in the *Senna tora* genome. (**a**) The number and distribution of the TDGs on the linkage groups. (**b**) The distribution of the synonymous substitutions per synonymous site calculated by the TDG pairs using the NG method; the peak values ranged from 0.10 to 0.14. (**c**) The KEGG pathway enrichment analysis of the *Senna tora* TDGs in 19 biochemical pathways, which were significantly enriched (*Padj* <0.05); the 19 pathway terms were divided into eight categories. (**d**) The family classification of 86 TDGs, which were significantly enriched into eight pathway terms of “biosynthesis of other secondary metabolites” category.

**Figure 2 ijms-24-04837-f002:**
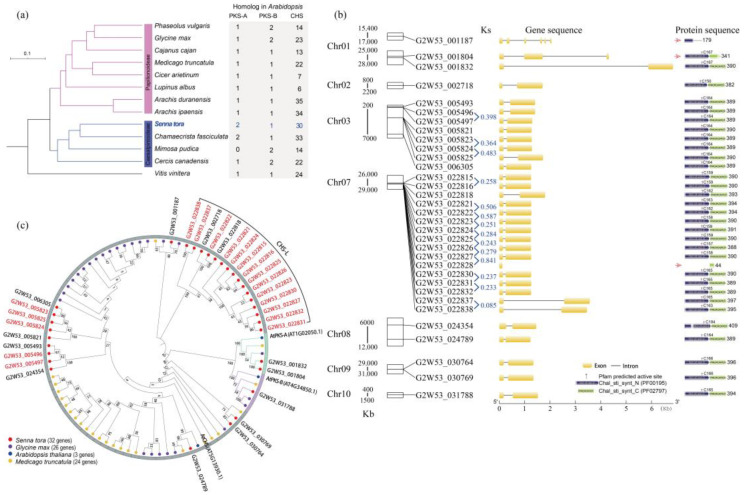
The genome-wide identification and analysis of the type III PKS superfamily genes. (**a**) The phylogenetic relationship and the number of type III PKS superfamily genes in *Senna tora* and other plant species in the Papilionoideae and Caesalpinoideae clades. (**b**) The distribution, structure, and domain of the type III PKS genes in the *S. tora* genome. The left side shows the linkage group or scaffold in which the gene is located, the gene ID and structure of type III PKS genes are displayed in the middle location, and the domains predicted by Pfam are given on the left side. (**c**) The unrooted phylogenetic tree shows the evolutionary relationships of the type III PKS protein sequences from *S. tora*, *Glycine max*, *Arabidopsis thaliana*, and *Medicago truncatula*. The gene IDs with red represent the TDGs in the *S. tora* genome.

**Figure 3 ijms-24-04837-f003:**
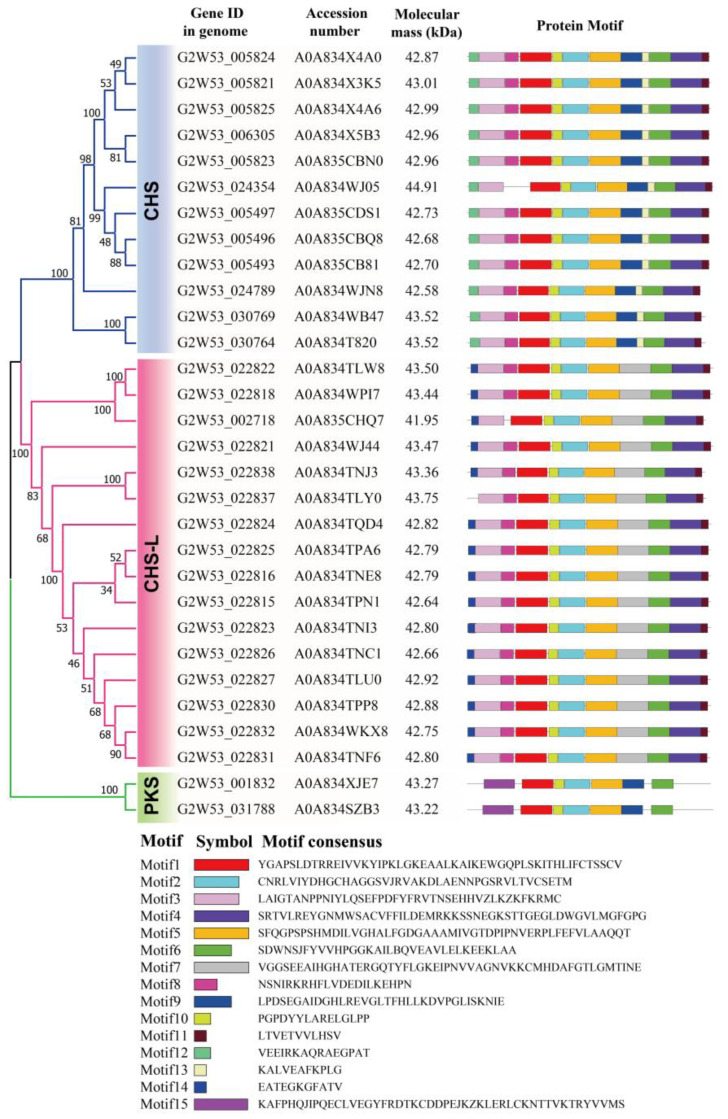
The protein phylogenetic and conserved motif analysis of the type III PKS superfamily members in *Senna tora*. The deduced amino acid sequences of the type III PKSs were obtained from UniProt (https://www.uniprot.org), accessed on 9 December 2022. The phylogenetic tree was constructed using the neighbor-joining method with 1000 bootstrap replicates in the MEGA7 program. The 15 conserved motifs were identified by MEME and are represented by different colored boxes.

**Figure 4 ijms-24-04837-f004:**
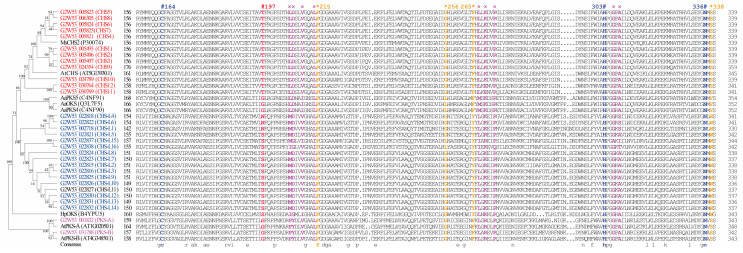
Comparison analysis of the deduced amino acid sequences of type III PKSs in *Senna tora* and other plant species. The first two letters of each gene indicate the abbreviated species name. Ms: *Medicago sativa*, At: *Arabidopsis thaliana*, Aa: *Aloe arborescens*, Hp: *Hypericum perforatum*. CHS: chalcone synthase, PKS: polyketide synthase, OKS: octaketide synthase, CHS-L: chalcone synthase-like. The catalytic triad (Cys164, His303 and Asn336) (in blue) and the critical active-site residue 197 (in red) are marked with #; the residues lining the active site (Phe215, Gly256, F265, and Ser338) (in yellow) are marked with * (numbering in *M. sativa* CHS), and the residues for the CoA binding (in purple) are marked with ×.

**Figure 5 ijms-24-04837-f005:**
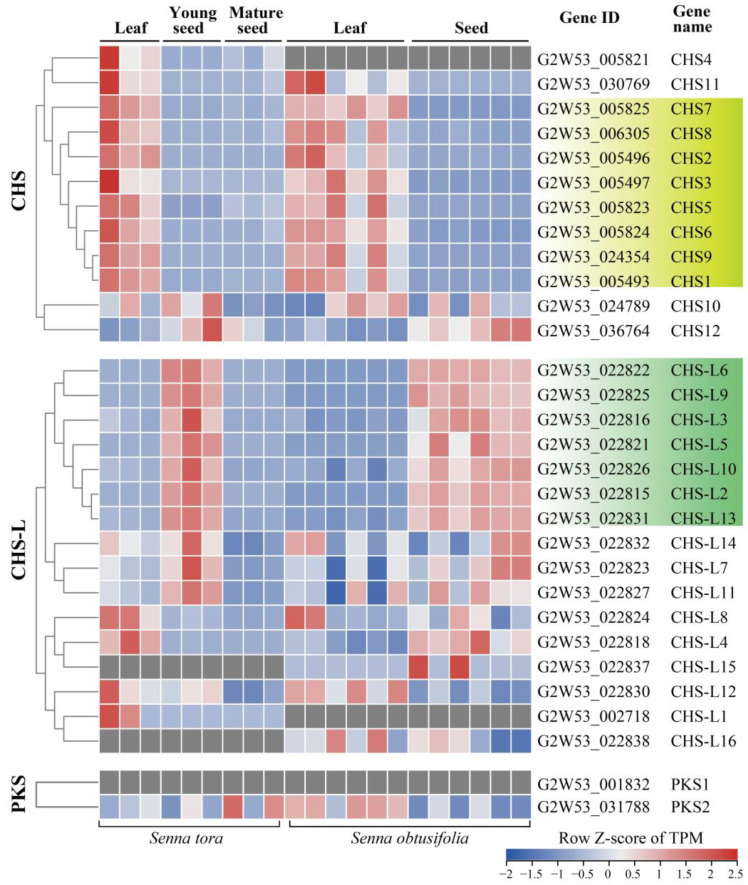
The expression profiles of the type III *PKS* genes in the *Senna tora* and *Senna obtusifolia* leaves and seeds. The relative expression levels of transcripts per million (TPM) were calculated from the transcriptome data (accession number SRP159435 and SRP144670). The color scale at the bottom right represents the TPM normalized by the row Z-score. The relatively low and high expression levels are indicated by blue and red, respectively.

**Figure 6 ijms-24-04837-f006:**
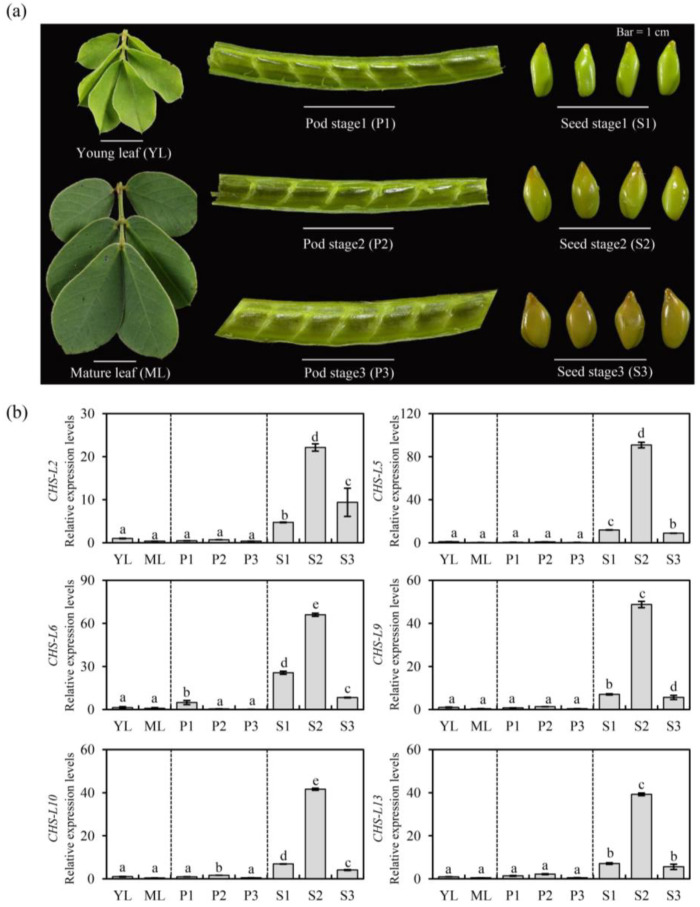
The expression patterns of six *CHS-L* genes in different *Senna tora* tissues. (**a**) The morphology of the leaves, pods, and seeds. Bars = 1 cm. (**b**) The relative expression levels of the six *CHS-L* genes were determined by quantitative real-time PCR (qRT-PCR). Each bar represents the means of three independent biological replicates with the standard error. Different letters demonstrate significant differences among the different tissues (*p* < 0.05).

**Figure 7 ijms-24-04837-f007:**
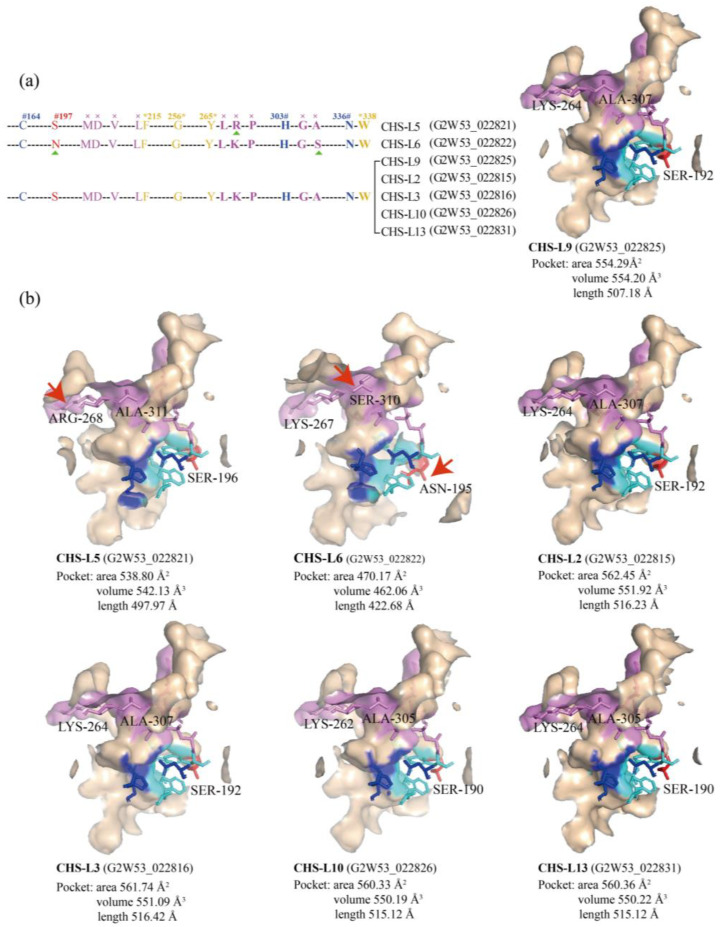
The analysis of the active-site residues and model structures of seven CHS-L proteins in *Senna tora*. (**a**) The catalytic active-site residues and CoA binding residues of six CHS-L proteins. The catalytic triad (Cys164, His303 and Asn336) (in blue) and the critical active-site residue 197 (in red) are marked with #; the residues lining the active site (Phe215, Gly256, F265, and Ser338) (in yellow) are marked with * (numbering in *M. sativa* CHS), and the residues for the CoA binding (in purple) are marked with ×. (**b**) Comparison of the 3D predicted protein structures of the seven CHS-L proteins. The active-site residues differing from other members are marked with red arrows.

## Data Availability

All relevant data are available from the corresponding author on request (luojiajia@catas.cn).

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
