# Peer review of "Genome-Wide Mining of the Tandem Duplicated Type III Polyketide Synthases and Their Expression, Structure Analysis of *Senna tora"

_ijms, 2023, doi:10.3390/ijms24054837_

Round 1
Reviewer 1 Report
This article presented Genome-wide mining tandem duplicated type III polyketide synthases and expression, structure analysis in Senna tora. The study is well organized and data is well arranged. The findings would be helpful for future studies. Before recommending this article for publication, there are some shortcomings for that should be resolve.
Methods are not well presented in the abstract. Which techniques were used for the analysis.
Also add quantitative and specific results in abstract section.
Provide economic, medicinal and industrial significance of Senna tora.
Why Senna tora was selected for this study should be mention in introduction.
Line 39 should be cited with relevant study. https://doi.org/10.1016/j.jep.2021.114515
Section 4.6 should be cited with recent study https://doi.org/10.1016/j.indcrop.2022.116090
Line 263-275 looks like results. The authors are directed to discuss the lines in the light of relevant literature based on obtained results.
Conclusion is well presented but some future recommendations should also be presented based on the obtained results.
Author Response
Response to Reviewer 1 Comments
This article presented Genome-wide mining tandem duplicated type III polyketide synthases and expression, structure analysis in Senna tora. The study is well organized and data is well arranged. The findings would be helpful for future studies. Before recommending this article for publication, there are some shortcomings for that should be resolve.
Point 1: Methods are not well presented in the abstract. Which techniques were used for the analysis.
Response 1: Thank you for your helpful suggestions. We agree with you, in the revised manuscript, we have modified the abstract. Please check this in details on “Abstract” section, lines 13-34 in the revised manuscript.
Point 2: Also add quantitative and specific results in abstract section.
Response 2: Thank you for your valuable suggestions. We have revised the abstract carefully. Please check this in details on lines 19-30 in the revised manuscript.
Point 3: Provide economic, medicinal and industrial significance of Senna tora.
Response 3: Thank you for your helpful suggestions. We agree with you, and we have added this to the “Introduction” section, please check this in details on lines 38-48 in the revised manuscript.
Point 4: Why Senna tora was selected for this study should be mentioned in introduction.
Response 4: Thank you for your valuable suggestions. We have made a major modification for the “Introduction” section. And we have explained that “Senna tora was considered as a model plant to research the biosynthetic pathways of anthraquinones, but the detailed regulation mechanisms of anthraquinone biosynthesis via the polyketide pathway in plants are largely unknown”, so that Senna tora was selected for this study. please check this in details on lines 39-72 in the revised manuscript.
Point 5: Line 39 should be cited with relevant study. https://doi.org/10.1016/j.jep.2021.114515
Response 5: Thank you for your helpful suggestions. We have added this study in line 51. Please check this in details on “Introduction” section.
Reference: 6. Zaman, W.; Ye, J.; Saqib, S.; Liu, Y.; Shan, Z.; Hao, D.; Chen, Z.; Xiao, P. Predicting potential medicinal plants with phyloge-netic topology: Inspiration from the research of traditional Chinese medicine. J. Ethnopharmacol. 2021, 281, 114515. http://doi.org/10.1016/j.jep.2021.114515.
Point 6: Section 4.6 should be cited with recent study. https://doi.org/10.1016/j.indcrop.2022.116090
Response 6: Thank you for your valuable suggestions. We have modified it. Please check this in details on line 394 in “Section 4.6”.
Reference: 33. Li, D.; Zaman, W.; Lu, J.; Niu, Q.; Zhang, X.; Ayaz, A.; Saqib, S.; Yang, B.; Zhang, J.; Zhao, H.; Lü, S. Natural lupeol level variation among castor accessions and the upregulation of lupeol synthesis in response to light. Ind. Crops Prod. 2023, 192, 116090. https://doi.org/10.1016/j.indcrop.2022.116090.
Point 7: Line 263-275 looks like results. The authors are directed to discuss the lines in the light of relevant literature based on obtained results.
Response 7: Thank you for your helpful suggestions. We agree with you. We have modified it, please check this in details on lines 276-284.
Point 8: Conclusion is well presented but some future recommendations should also be presented based on the obtained results.
Response 8: Thank you for your valuable suggestions. We agree with you. And we have made a major modification for the “Conclusion” section, please check this in details on lines 412-425.
Reviewer 2 Report
To,
The Editor,
IJMS, MDPI,
Manuscript ID: ijms-2225627
Subject: Submission of comments of the manuscript in “IJMS"
Dear Editor IJMS, MDPI,
Thank you very much for the invitation to consider a potential reviewer for the manuscript (ID: ijms-2225627). My comments responses are furnished below as per each reviewer’s comments.
In the reviewed manuscript, the authors identified 3,087 TDG. Ks analysis indicated that TDGs had recently undergone duplication. KEGG enrichment analysis revealed that type III polyketide synthases were the most enriched TDGs, as evidenced by 14 tandem duplications out of 16 CHS-L genes. Expression profiles displayed that tandem duplicated CHS-L2/3/5/6/9/10/13 genes were mainly expressed in seeds. The key active-site residues and three-dimensional models of CHS-L2/3/5/6/9/10/13 proteins showed similarity. These findings suggest that the recent tandem duplication of CHS-L genes might play important roles in the biosynthesis of rich anthraquinones in S. tora seeds, and the seven CHS-L2/3/5/6/9/10/13 genes would provide candidate genes for further research.. There are some minor and major comments. The author should revise as per my comments and suggestions.
- I have read the entire manuscript and my initial comment is that the manuscript is poorly written. I have significant concerns about the grammar and vocabulary of the manuscript; therefore, I recommend the authors to use an English proofreading service.
- The abstract does not reflect the whole story, revise it
- The keywords must be in alphabetical order.
- The writing style of the paper is very poor. There are many grammatical mistakes. Long sentences with noticeable grammatical mistakes are frequently present throughout the manuscript. There are many typos mistakes in this whole manuscript. The author should check the whole manuscript.
- The introduction part is not impressive and systematic. In the introduction part, the authors should elaborate on the scientific issues in plant research. The Content of the introduction is effective in essence but very poorly presented, significant improvements are needed in presenting the proper background of the work undertaken
- The figures are quite low resolution and difficult to make out. Higher-resolution versions will be needed for publication. Further, the text in the figures is not readable, for example, in Figures 1, 2, 3, 4, 6b, and 7.
- The qRT-PCR methodology provided is also very vague and confusing. Please provide more details like what was the calibrator used in the study. I assume the authors have used the control as the calibrator. If so, the authors should not include the control within the bar graph as it represents the fold change between the treated vs control and a fold change of “1” for the ‘control’ doesn’t make any sense. Also, would be good to provide details on what reagents (details of probes used, if any, if SYBR was used then details for that, etc.) and real-time PCR machines were used in the current study.
- The discussion should be interpreted with the results as well as discussed in relation to the present literature.
- The conclusion section is very short. The author should emphasize this in a better way.
Author Response
Response to Reviewer 2 Comments
Dear Editor IJMS, MDPI,
Thank you very much for the invitation to consider a potential reviewer for the manuscript (ID: ijms-2225627). My comments responses are furnished below as per each reviewer’s comments.
In the reviewed manuscript, the authors identified 3,087 TDG. Ks analysis indicated that TDGs had recently undergone duplication. KEGG enrichment analysis revealed that type III polyketide synthases were the most enriched TDGs, as evidenced by 14 tandem duplications out of 16 CHS-L genes. Expression profiles displayed that tandem duplicated CHS-L2/3/5/6/9/10/13 genes were mainly expressed in seeds. The key active-site residues and three-dimensional models of CHS-L2/3/5/6/9/10/13 proteins showed similarity. These findings suggest that the recent tandem duplication of CHS-L genes might play important roles in the biosynthesis of rich anthraquinones in S. tora seeds, and the seven CHS-L2/3/5/6/9/10/13 genes would provide candidate genes for further research. There are some minor and major comments. The author should revise as per my comments and suggestions.
Point 1: I have read the entire manuscript and my initial comment is that the manuscript is poorly written. I have significant concerns about the grammar and vocabulary of the manuscript; therefore, I recommend the authors to use an English proofreading service.
Response 1: Thank you for your valuable suggestions. The manuscript has been ‘spell checked’ and ‘grammar checked’ by MDPI (https://www.mdpi.com/authors/english.), and we also have carefully revised the full text.
Point 2: The abstract does not reflect the whole story, revise it
Response 2: Thank you for your helpful suggestions. We agree with you. We have made a major modification for the “Abstract” section, please check this in details on lines 13-34.
Point 3: The keywords must be in alphabetical order.
Response 3: Thank you for your valuable suggestions. We have modified it, please check this in details on line 35, 36.
Point 4: The writing style of the paper is very poor. There are many grammatical mistakes. Long sentences with noticeable grammatical mistakes are frequently present throughout the manuscript. There are many typos mistakes in this whole manuscript. The author should check the whole manuscript.
Response 4: Thank you for your helpful suggestions. It is our fault. Now, The manuscript has been ‘spell checked’ and ‘grammar checked’ by MDPI (https://www.mdpi.com/authors/english.), and we also have carefully revised the full text.
Point 5: The introduction part is not impressive and systematic. In the introduction part, the authors should elaborate on the scientific issues in plant research. The Content of the introduction is effective in essence but very poorly presented, significant improvements are needed in presenting the proper background of the work undertaken
Response 5: Thank you for your valuable suggestions. We have made a major modification for the “Introduction” section. Please check this in details on lines 39-98 in the revised manuscript.
Point 6: The figures are quite low resolution and difficult to make out. Higher-resolution versions will be needed for publication. Further, the text in the figures is not readable, for example, in Figures 1, 2, 3, 4, 6b, and 7.
Response 6: Thank you for your helpful suggestions. We are so sorry. Now, we have re-uploaded a vectorgraph in PDF format, and we re-edited the text in these figures 1, 2, 3, 4, 6b, and 7.
Point 7: The qRT-PCR methodology provided is also very vague and confusing. Please provide more details like what was the calibrator used in the study. I assume the authors have used the control as the calibrator. If so, the authors should not include the control within the bar graph as it represents the fold change between the treated vs control and a fold change of “1” for the ‘control’ doesn’t make any sense. Also, would be good to provide details on what reagents (details of probes used, if any, if SYBR was used then details for that, etc.) and real-time PCR machines were used in the current study.
Response 7: Thank you for your valuable suggestions. We have modified the “qRT-PCR methodology”, added the details of reagents and real-time PCR machines etc. In addition, we evaluated the relative expression levels of CHS-L genes in different tissues (leaves, pods and seeds) using PTB gene as the internal control. The bar graphs showed the relative expression levels of CHS-L genes versus PTB gene, calculated by the 2-ΔΔCT method, not fold change among the different tissues. Please check this in details on lines 380-395.
Point 8: The discussion should be interpreted with the results as well as discussed in relation to the present literature.
Response 8: Thank you for your helpful suggestions. We have made a major modification for the “3. Disscution” section, please check this in details on lines 251-313.
Point 9: The conclusion section is very short. The author should emphasize this in a better way.
Response 9: Thank you for your valuable suggestions. We have made a major modification for the “5. Conclusion” section, please heck this in details on lines 412-425.
Reviewer 3 Report
This study demonstrates that tandem duplication of chalcone synthase-like genes might play important roles in the biosynthesis of anthraquinones in S. tora seeds, on the basis of gene duplication analysis of type III polyketide 2 synthases and their expression in
in Senna tora. The three-dimensional models were also built and expression levels f these CHS-L genes were determined. Motifs analysis of type III polyketide synthases and active site analysis were performed. The study suggests that chalcone synthase-like genes recently expanded by tandem duplication, in response to the biosynthesis of diverse anthraquinone metabolites in Senna.
My main doubt is whether this manuscript fits the profile of the journal since it is devoted rather to gene evolution and protein structure. However, since chalcone synthases are so important for the synthesis of important plant antioxidants, the manuscript can be considered for publication in this journal, especially in the Special Issue.
Remarks:
Line12: “food homologous crops’, please reformulate
Lines 14/15: “the roles of CHS-L members in S. tora remain less definitive”, please reformulate
Line 412: “suggested”, please change to “suggests”
Author Response
Response to Reviewer 3 Comments
This study demonstrates that tandem duplication of chalcone synthase-like genes might play important roles in the biosynthesis of anthraquinones in S. tora seeds, on the basis of gene duplication analysis of type III polyketide 2 synthases and their expression in Senna tora. The three-dimensional models were also built and expression levels of these CHS-L genes were determined. Motifs analysis of type III polyketide synthases and active site analysis were performed. The study suggests that chalcone synthase-like genes recently expanded by tandem duplication, in response to the biosynthesis of diverse anthraquinone metabolites in Senna.
My main doubt is whether this manuscript fits the profile of the journal since it is devoted rather to gene evolution and protein structure. However, since chalcone synthases are so important for the synthesis of important plant antioxidants, the manuscript can be considered for publication in this journal, especially in the Special Issue.
Remarks:
Point 1: Line12: “food homologous crops’, please reformulate
Response 1: Thank you for your helpful suggestions. We have revised it as “Senna tora is one of the homologous crops used as a medicinal food containing an abundance of anthraquinones.” Please check this in details on line 13 in revised manuscript.
Point 2: Lines 14/15: “the roles of CHS-L members in S. tora remain less definitive”, please reformulate
Response 2: Thank you for your valuable suggestions. We have modified it. Please check this in details on lines 17,18 in the revised manuscript.
Point 3: Line 412: “suggested”, please change to “suggests”
Response 3: Thank you for your helpful suggestions. We agree with you and we have changed it, please check this in details on line 421 in the revised manuscript.
Round 2
Reviewer 2 Report
Dear Editor,
Thank you for providing the opportunity to review the revised manuscript. The manuscript is improved considerably after revision according to the reviewer's comment. Now this study is a suitable contribution to the IJMS. I recommend the manuscript for publication.
Thank you
With best regards